# Investigation of Urea Uniformity with Different Types of Urea Injectors in an SCR System

**Muhammad Khristamto Aditya Wardana [1,2], Kwangchul Oh [3]** 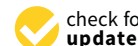 **and Ocktaeck Lim [4,*]**

[1] Graduate School of Mechanical Engineering, University of Ulsan, Ulsan 680-749, Korea; m.k.aditya.w@gmail.com

[2] Research Centre of Mechatronic and Electrical Power, Indonesian Institute of Sciences, Jl Sangkuriang Komplek LIPI Gd 20 Cisitu, Bandung, Jawa Barat 40135, Indonesia

[3] Korea Automotive Technology Institute, 303 Poongse-ro, Poongse-myun, Dongnam-gu, Chenahn 31214, Korea; kcoh@katech.re.kr

[4] School of Mechanical Engineering, University of Ulsan, Ulsan 44610, Korea

[*] Correspondence: otlim@ulsan.ac.kr; Tel.: +82-52-259-2852

**Abstract:** Heavy-duty diesel engines in highway use account for more than 40% of total particulate and nitrogen oxide (NO$x$) emissions around the world. Selective catalytic reduction (SCR) is a method with effective results to reduce this problem. This research deals with problems in the urea evaporation process and ammonia gas distribution in an SCR system. The studied system used two types of urea injectors to elucidate the quality of ammonia uniformity in the SCR system, and a 12,000-cc heavy-duty diesel engine was used for experimentation to reduce NO$x$ in the system. The uniformity of the generated quantities of ammonia was sampled at the catalyst inlet using a gas sensor. The ammonia samples from the two types of urea injectors were compared in experimental and simulation results, where the simulation conditions were based on experimental parameters and were performed using the commercial CFD (computational fluid dynamics) code of STAR-CCM+. This study produces temperatures of 371 to 374 °C to assist the vaporization phenomena of two injectors, the gas pattern informs the distributions of ammonia in the system, and the high ammonia quantity from the I-type urea injector and high quality of ammonia uniformity from the L-type urea injector can produce different results for NO$x$ reduction efficiency quality after the catalyst process. The investigations showed the performance of two types of injectors and catalysts in the SCR system in a heavy-duty diesel engine.

**Keywords:** emissions; ammonia; selective catalytic reduction (SCR); urea injector; heavy-duty diesel engine; urea water solution (UWS)

## 1. Introduction

The heavy-duty diesel engine produces significant nitrogen oxide (NO$x$) and particulate matter (PM) emissions that negatively affect human health [1–5]. For this reason, the global automotive industry is required to reduce NO$x$ emissions from their products, and it is anticipated that future standards will require exhaust after-treatment research to solve the NO$x$ emission problem [2]. To meet these standards, the combination of diesel particulate filters (DPFs) to control PM emissions and selective catalytic reduction (SCR) to reduce NO$x$ emissions have been used. An SCR system with particles of ammonia can reduce NO$x$ emissions by more than 80% [6]. In that study, a water solution of urea was injected into the exhaust system of a diesel engine, and the hot exhaust gas evaporated the urea and generated ammonia to control NO$x$ emissions.

However, SCR systems need more improvements to reach the optimal NO$x$ reduction efficiency. Many factors make it difficult to achieve this goal, the main problem being the mixing process between

ammonia and NO*x* emissions [6–10]. If the urea decomposition process is incomplete, solids will be deposited in the system. The amount of solid deposited will also hamper the distribution of ammonia and gas. Finally, the deposited solid was grown and lowered production of ammonia in the system. That reaction also effects to the quality of NO*x* reduction efficiency. The difference in the quantity of ammonia and NO*x* could lower NO*x* reduction quality in the system [11]. Based on this assumption, the urea evaporation process is the important indicator to elucidate the urea decomposition process and solid deposition in the SCR system. To achieve good quality of evaporation, the urea injector distribution and quantity of gas temperature will be analyzed in this study.

Our experiments involved two types of urea injectors for a heavy-duty diesel engine—a Hyundai D6CC, operated at 1000 rpm, the highest possible performance. Our simulations used the commercial computational fluid dynamics (CFD) code of STAR-CCM+ version 11.04, with which the flow factors, temperature factors, and urea decomposition factors, which can affect the quality of the urea decomposition process in the system [6–10,12], were thoroughly examined.

The two types of urea injectors used in this study determined the shape of the urea injection in the SCR system. The performance of urea injection will improve the quality of the urea decomposition process and prevent solid deposition. The quantities of exhaust gas and urea injection in the simulations were the same as those in the experiments. The ammonia homogenization from the two types of injectors was investigated at the inner part of the catalyst surface with a 19-point gas sensor. The quality and the quantity of ammonia can increase the NO*x* reduction quality of the heavy-duty diesel engine. The quality of the catalyst and the NO*x* efficiency were determined by a gas emission analyzer to determine the NO*x* quantity from this system.

## 2. Results and Discussion

### 2.1. UWS Distribution Gas

Previous experimental studies with heavy-duty diesel engine SCR systems lack sufficient information to understand the urea distribution process, but our STAR-CCM+ simulations clearly showed the phenomena inside the chosen SCR system. Figure 1 shows the urea distribution for the I-type and L-type injectors in that SCR system. According to [6], the impacts of strong gas flow from an engine can improve ammonia homogenization in its SCR system.

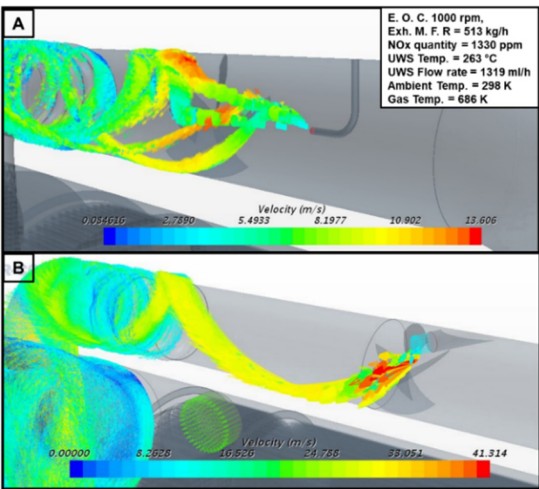

**Figure 1.** Urea distribution phenomena in the SCR system: (**A**) L-type urea injector and (**B**) I-type urea injector.

In this study, urea was distributed to the system by strong exhaust gas. The urea water solution consisted of ~60% water ($H_2O$) and ~40% urea. The $H_2O$ evaporated before the urea; this reaction occurred because water is lighter than urea, and it also assisted the urea evaporation process in the

system. The evaporation phenomena reached higher values near the SCR catalyst because of the high pressure of the gas. Moreover, the SCR catalyst in this study had small porosity (>50 nm) and hampered the distribution of gas; these phenomena also improved the mixing process between ammonia and NO$x$ before entering the SCR catalyst. The small porosity in the catalyst also assisted the SCR system to produce high-quality NO$x$ reduction efficiency.

Figure 2 shows the temperature phenomena inside the SCR system. Koebel and Strutz [13] explained the thermal and hydrolytic decomposition processes of urea in the 150 to 500 °C temperature range. Although the usual temperature in commercial SCR systems is around 120 to 350 °C, their theory might elucidate the decomposition process in heavy-duty diesel engines. Our simulation used a 12,000-cc heavy-duty diesel engine operating at 1000 rpm, which could produce temperatures of 371 to 374 °C to assist vaporization of urea in the SCR system.

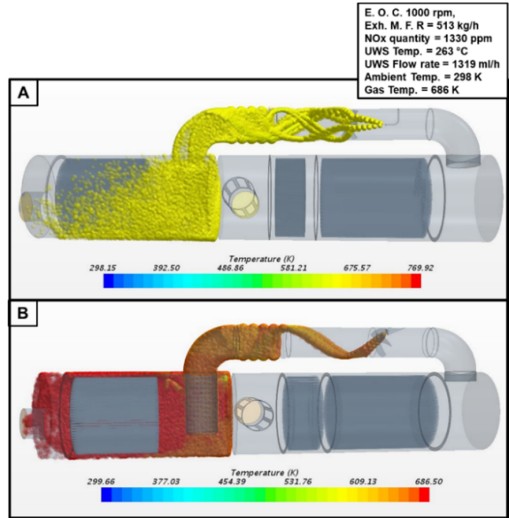

**Figure 2.** Temperature phenomena in the SCR: (**A**) L-type urea injector and (**B**) I-type urea injector.

The urea distribution is also affected by the quantity of exhaust gas from the engine. At 1000 rpm, that engine could produce 513 kg/h of exhaust mass and 1083 g/h of NO$x$ mass. That gas quantity can assist in the distribution of urea to all systems, although the higher exhaust mass inflicts a higher NO$x$ conversion process in the SCR system. Figure 3 shows the urea distribution pattern in the SCR system with the engine running at 1000 rpm. The urea distribution pattern can clearly explain the quality of the ammonia generation process in the system. This process and phenomena will be more thoroughly explained by the experiment results in the next section.

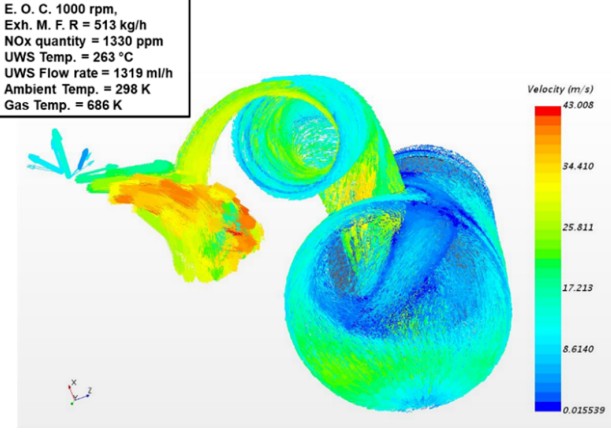

**Figure 3.** The urea distribution pattern in the SCR system.

## 2.2. UWS Evaporation Process

The distribution gas in this study was described in the previous section. The quality of the urea process is reflected in the quantity of solid depositions that occur in the system. In commercial SCR systems, the $H_2O$ and urea are usually difficult to evaporate [1,6,14]: The urea particles easily attach to and settle on the system walls, and this process is the main source of solid deposits in SCR systems. However, this study found that urea was evaporated well. Figure 4 shows the lower quantity of mass flux slip in the system, which demonstrates that the gas particles were distributed to the catalyst with hardly any urea particles becoming deposits in the system. This was possible because mass flux simulations can identify any abnormal distributions of gas particles in a system.

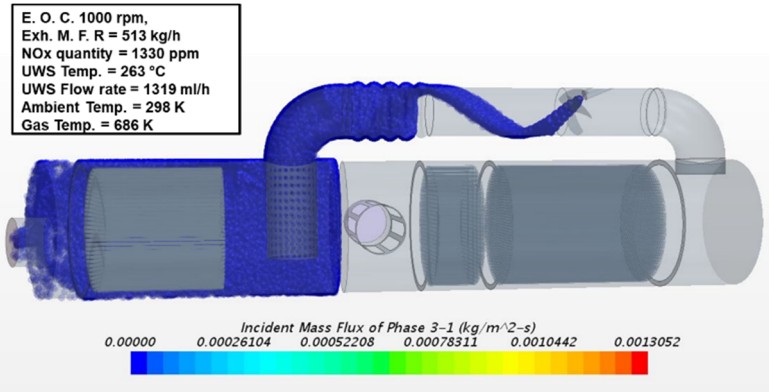

**Figure 4.** The quantity of incident mass flux in the SCR system.

The quality of the urea process also can be described with the NO*x* reduction efficiency derived from the experimental results. The quality of the urea evaporation process is reflected by the NO*x* concentration in the outlet catalyst, as determined in this study with the Horiba MEXA-7100 DEGR emission analyzer, as shown in Figure 5. The I-type urea injector used in this system produced 5% more ammonia than the L-type injector. This value supports our simulation results in the previous section that suggest that I-type injectors are better than L-type injectors in terms of improving the ammonia generation process.

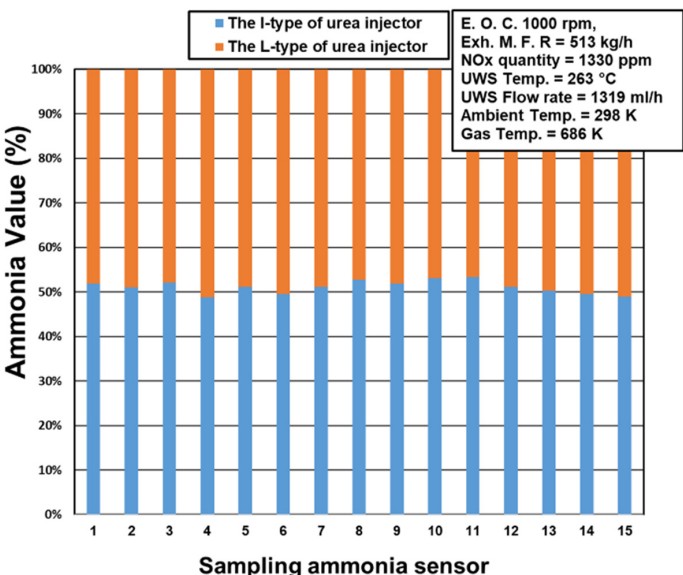

**Figure 5.** The ammonia distribution values at the catalyst inlet.

### 2.3. Ammonia Uniformity

The distribution of gas and the quality of produced urea are clearly described in the previous section. Figure 6 shows the ammonia uniformity quality from the two types of urea injectors. The simulation results from previous section showed that the I-type urea injector produced more ammonia than the L-type injector. However, the distribution of ammonia gas is imperfect, like L-type injector. The I-type injector had better vaporization and saturation, as can be seen from the ammonia patterns in Figure 6. The value of the gas was considerably similar for the two injector types, but the gas flow and quality of ammonia were different. The pattern of ammonia uniformity at the catalyst inlet can explain the gas distribution inside the SCR system. Although the simulation showed that the I-type urea injector produced higher ammonia quantity than the L-type injector, the L-type injector produced a higher distribution of the gas than I-type injector. This result can be observed in Figure 6—the catalyst inlet showed that the L-type injector is more uniform than the I-type injector. To validate the ammonia uniformity from the simulation, experiments with the commercial SCR system in the Hyundai D6CC were conducted.

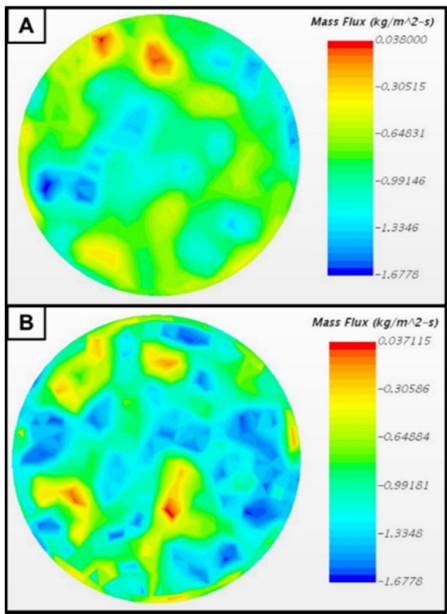

**Figure 6.** Simulation results of ammonia uniformity in the catalyst inlet: (**A**) L-type urea injector and (**B**) I-type urea injector.

Figure 7 shows the sample catalyst with 19 gas sensors to measure the quantity of ammonia particles in the system, a highly effective way to describe ammonia patterns in the experiments. The number of ammonia particles from each sensor was computed in the gas analyzer, which also showed the values from each sensor. Thus, ammonia uniformity can be described by the color contrast according to the sensor values.

The results from the gas analyzer are given in Figure 8, showing ammonia uniformity for the I-type and L-type urea injectors in the system. In terms of the pattern and quality of ammonia uniformity, the L-type urea injector was better than the I-type injector, meaning that the results from the simulation agreed well with the ammonia distribution gas from the experimental results. Accordingly, the I-type injector produced more ammonia but with less uniformity at the catalyst inlet, while the L-type injector produced less ammonia but with better uniformity at the catalyst inlet. Thus, the urea vaporization process and the ammonia distribution gas can affect the efficacy of an SCR system. Although this work compared only two types of urea injectors with a standard mixer fan position (front position), our results can be useful for improving commercial SCR systems in heavy-duty diesel engines with the same specifications.

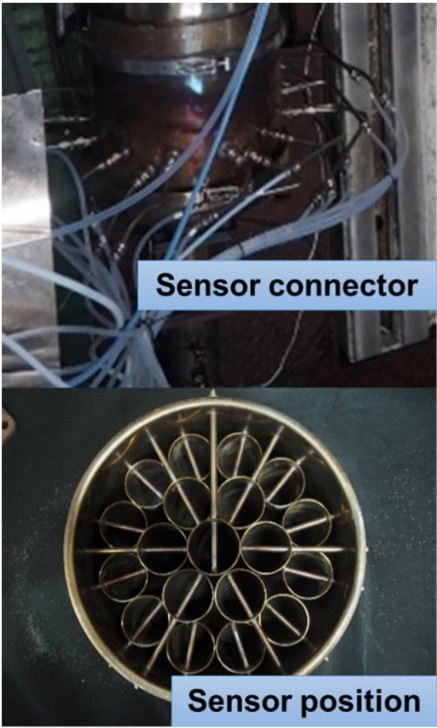

**Figure 7.** The positions of 19 sensors at the catalyst inlet for the analysis of ammonia uniformity.

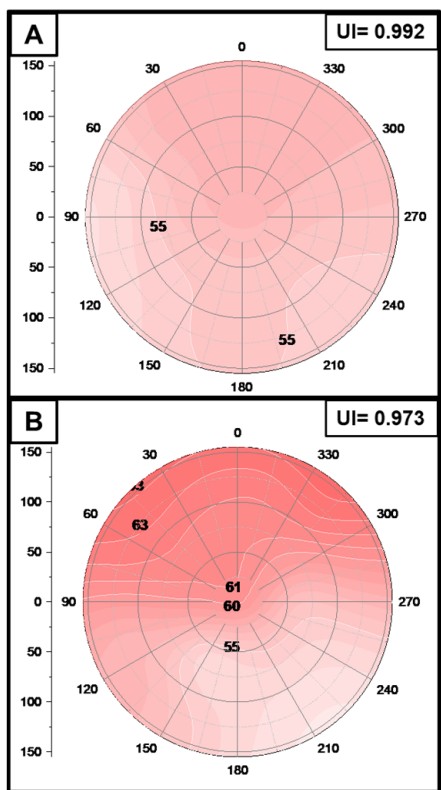

**Figure 8.** Experimental results of ammonia uniformity in the catalyst inlet: (**A**) L-type urea injector and (**B**) I-type urea injector.

*2.4. NOx Reduction Efficiency*

The ammonia quantity, ammonia uniformity and catalyst substrate in the SCR system were indicators to identify high-quality of NOx efficiency. NOx from the engine will mix with ammonia in the system; if the quantity of ammonia gas lower is than NOx, the amount of NOx will remain high after passing from the catalyst and produce low NOx efficiency in the SCR system. The low NOx efficiency can also be found if the quality of ammonia uniformity is low. The ammonia is not distributed perfectly in the system and produces an uncompleted mixing process of ammonia and NOx. Although all the processes are filtered by the catalyst, the ammonia quantity and ammonia uniformity are still important in this system.

Figure 9 show the experiment result with 20 test samples with the gas analyzer at the catalyst outlet (Horiba MEXA-7100 DEGR). The results show the quality of NOx efficiency from the I-type urea injector and the L-type urea injector. Although differences in the results from both of the injectors are not big, it is clear that the I-type urea injector provides higher NOx reduction efficiency than the L-type urea injector. Based on this information, it can be concluded that the I-type injector has high ammonia quantity, which results in providing assistance to the mixing process in the system to produce high NOx efficiency. However, these results still require improvements, because 60% of the NOx reduction efficiency after the catalyst filter process remains an average result. The increasing ammonia quantity and quality are the main areas to be explored in future plans regarding the SCR system.

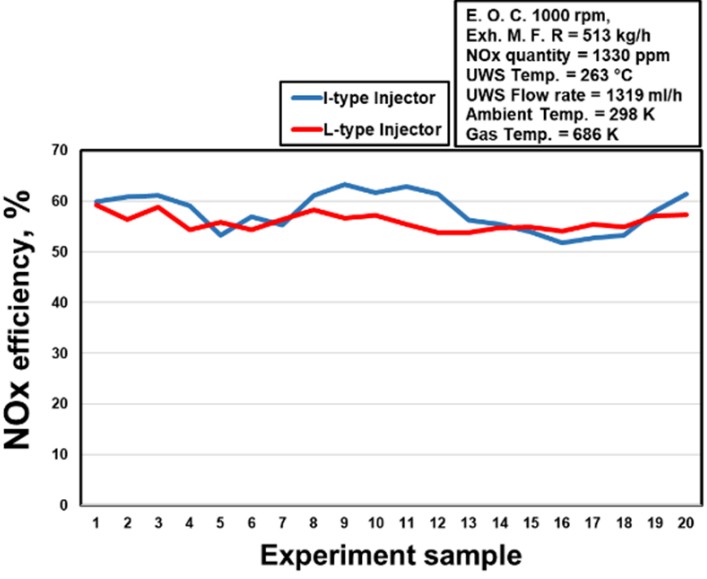

**Figure 9.** The experiment result of NOx efficiency.

## 3. The Simulation Model and the Geometry Condition

For reliable prediction of urea injection shape and temperature distribution in real applications of the studied SCR system, a kinetic model for urea distribution and decomposition was integrated into the CFD model using STAR-CCM+. In an interesting study of turbulence models in SCR systems [6], two CFD models to investigate urea uniformity and NOx reduction efficiency in SCR systems were used. The urea injector for their simulation of optical access had a 90° angle relative to the main flow; the simulation clearly showed urea decomposition and gas concentration in the system. Their second simulation used a commercial SCR system in a Mercedes-Benz ML350, in which the urea injector was released at a 15° angle to the system. This position was based on the experimental system used by the authors.

Our current simulation studies were performed on the commercial SCR used in a Hyundai D6CC diesel engine. However, this study used two types of urea injectors in the system. Figure 10 explains

the SCR system geometry in the experiments, and Figure 11 explains the simulation model. The first simulation used an L-type urea injector (the original type) with a 90° angle to the main flow, and the second simulation used an I-type urea injector (the improvement) with a 32° angle to the system. The two types of urea injectors were analyzed to obtain optimal ammonia homogenization based on decomposition and evaporation phenomena in the system.

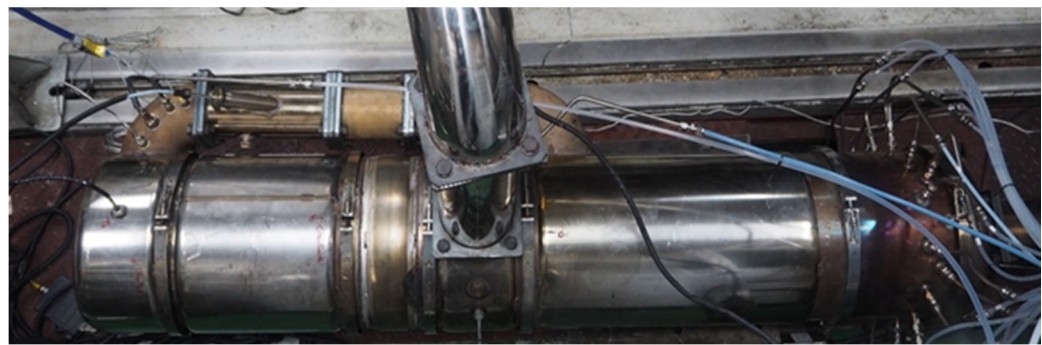

**Figure 10.** Selective catalytic reduction (SCR) system geometry in a Hyundai D6CC heavy-duty diesel engine.

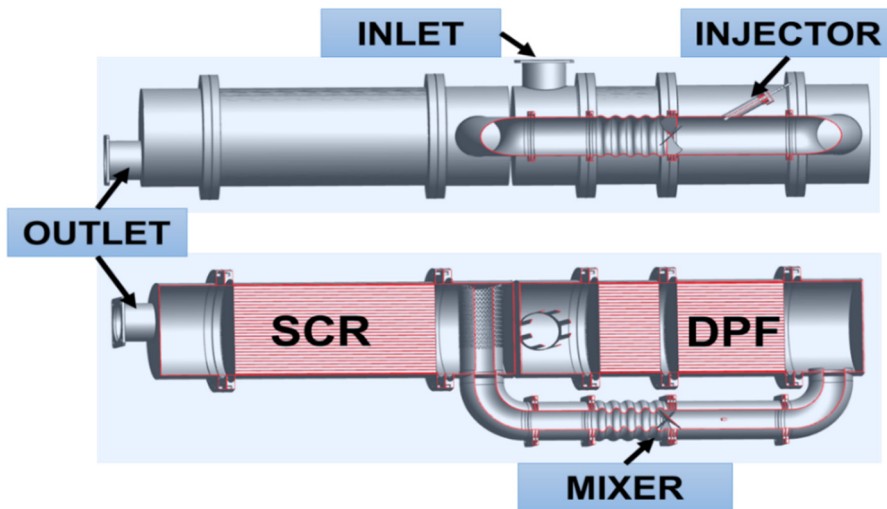

**Figure 11.** Simulation geometry of the commercial SCR system in the D6CC.

The two types of urea injectors are shown in Figures 12 and 13. The simulations were of different injector types, but the injector holes and performance of the urea water solution (UWS) parameter were similar. The dimensions and sizes followed those of the D66CC: each urea injector had three spray holes with a 120 µm diameter, each with a 7° cone angle, and could produce a urea mass flow rate of $8.05 \times 10^5$ kg/s. The urea injection mechanism process is shown in Figure 14 [6]. The production of ammonia and CO in this system was also observed as a result of the reaction between water vapor and urea. Khristamto et al. [6] and Koebel et al. [8] suggested that the decomposition into ammonia and isocyanic acid is dominant.

$$NH_2 - CO - NH_2 \ \rightarrow \ HNCO + NH_3 \tag{1}$$

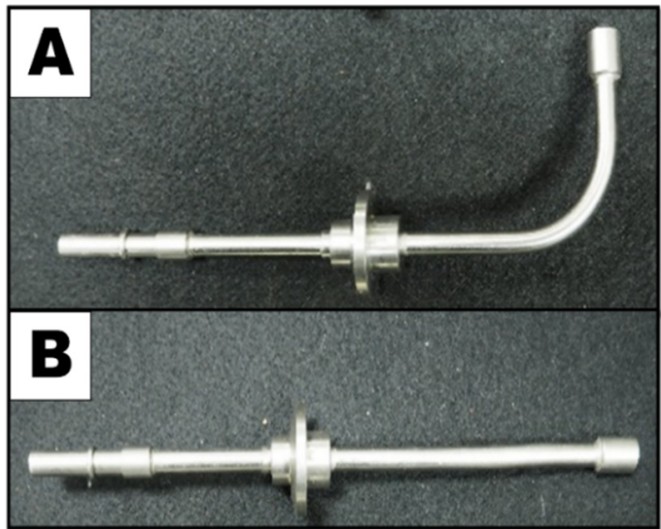

**Figure 12.** The urea injectors: (**A**) L-type (original type) and (**B**) I-type (improved type).

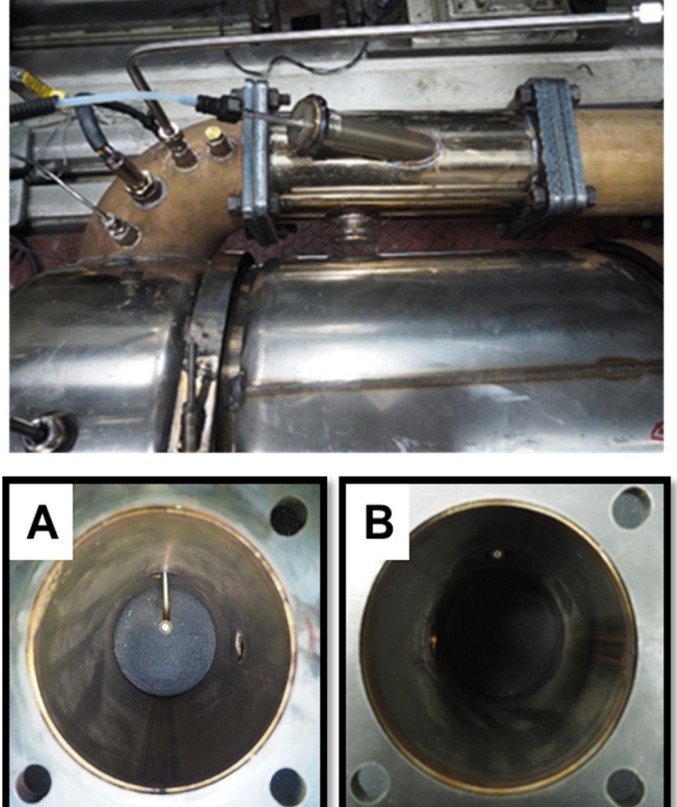

**Figure 13.** The urea injectors inside the SCR system: (**A**) L-type (original type) and (**B**) I-type (improved type).

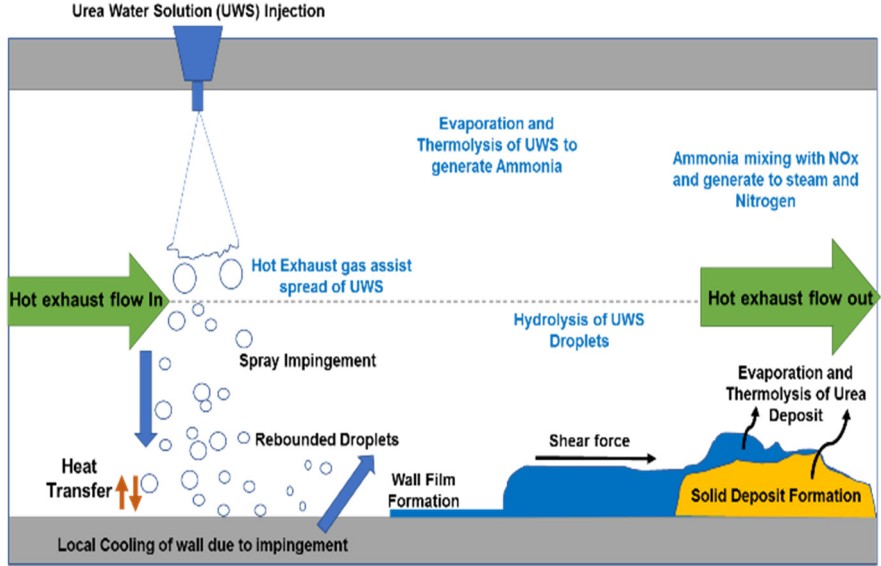

**Figure 14.** Visualization of the urea injection mechanism process.

Isocyanic acid could hydrolyze with water and produce carbon dioxide and ammonia in the SCR reaction mechanism [6]:

$$HNCO + H_2O \rightarrow CO_2 + NH_3 \tag{2}$$

Urea could also hydrolyze with water to produce carbon dioxide and ammonia [6]:

$$NH_2 - CO - NH_2 + H_2O \rightarrow 2NH_3 + CO_2 \tag{3}$$

The amounts of ammonia and isocyanic acid introduced into the catalyst were similar, and the amount of urea was observed under their experimental conditions [13]. In this study, the UWS is described with an evaporative approach to urea water solution (~60% water ($H_2O$) and ~40% urea); the concentrations of fluid masses are defined in each computational cell to represent the mass distribution in the SCR system.

The physics modeled in STAR-CCM+ comprised a Euler–Lagrange approach for gas flow and UWS injection, spray/wall interaction, liquid film formation, and evaporation. For the gas phase, a RANS (Reynolds-averaged Navier–Stokes) approach was used along with a k-epsilon turbulence model. Fischer and Simon [12] claimed that the Reynolds' stress model is better for computing anisotropic characteristics to understand the turbulence in the swirl flow. In this study, the simulation had a strong swirl flow to assist the mixing performance in the SCR system. The RANS model in STAR-CCM+ could well predict the noticeable flow between the primary swirl core and the outer secondary vortices. The turbulent kinetic energy from the system (*k*) will replace the Reynolds stresses model, as follows:

$$k = \sum_{i=1}^{3} \frac{\overline{U_l^l U_l^l}}{2}. \tag{4}$$

The RANS model was used to compute the complete tensor in the SCR system. Equations (5) and (6) show the calculation of turbulence intensity in the SCR system $\overline{U_l^l U_l^l}$ with the velocity component (*u*) in the *i*-direction as follows:

$$\overline{u_l} = \frac{1}{N} \sum_k u_{k,i} \tag{5}$$

$$\overline{U_l^l U_l^l} = \frac{1}{N} \sum_k \left( U_{k,i}^2 - \overline{u_i^2} \right) \tag{6}$$

In this study, the fluid film model solved the energy transport equations, species of gas, mass flow rate, momentum, and volume fraction.

$$\frac{d}{dt}\int_V \rho_f dV + \int_A \rho_f U_f.\, da = \int_V \frac{S_u}{h_f} dV \tag{7}$$

In this particular case, ($V$) is the volume and ($A$) is the area, which are functions of film thickness and its distribution. $U_f$ represents film velocity, $\rho_f$ represents film density, and $S_u$ is the mass source per area. The STAR-CCM+ software provided the equation models for evaporation of liquid film into gas phase and condensation from gas phase to liquid film. The species mass flow of each component was obtained at the interface between gas and film, resulting in the following conservation equation:

$$\rho Y_i(u-h) - \rho D_i \frac{dY_i}{dy} = \rho_f D_{f,i}\left(u_f - h\right) - \rho_f D_{f,i} \frac{dY_i}{dy}\bigg|_f \tag{8}$$

where $Y_i$ represents the gas mass fraction, $u$ represents the velocity of the gas, $D_i$ is the molar diffusion coefficient, and $h$ is the film thickness. $y$ in this equation represents the direction of the gas to the wall. For a nonmoving liquid ($u_f = 0$), the evaporation rate can be expressed by

$$\left[1 - \sum_j^{N_L} Y_j\right] \dot{m}_{vap} = -\sum_j^{N_L} \rho D_i \frac{dY_i}{dy} \tag{9}$$

where $N_L$ represents the number of components in the liquid film. The interface of the mass fraction ($Y_j$) is required to calculate the evaporation rate in the SCR system, where $\gamma i \approx 1$. The boundary conditions used in this study are shown in Table 1.

**Table 1.** Boundary conditions and turbulence models.

| Setup | Turbulence Model | Boundary Condition (P = Pressure) (V = Velocity) (T = Temperature) (I = Inlet) (O = Outlet) |
|:---:|:---:|:---:|
| 1 | RANS Model | PI-VO |

The quality of ammonia uniformity was defined based on the ammonia mass flux quantity in the system. The ammonia mass flux also could describe the performance of SCR system to reduce the quantity of NOx. The analysis of ammonia utilization in the system is as follows:

$$\gamma_{\text{mass\_flux}} = 1 - \frac{\int_A \left( m''_{NH3} - \overline{m}''_{NH3} \right) dA}{2\overline{m}''_{NH3}} \tag{10}$$

$$NOx_{\text{Conversion value}} = \left(1 - \frac{NOx \text{ Outlet}}{NOx \text{ inlet}}\right) \times 100\% \tag{11}$$

where $m''_{NH3}$ represent the ammonia mass flux quantity on the plane position, $\overline{m}''_{NH3}$ is the quality of the ammonia mass flux, and A is the surface area in the SCR system (catalyst inlet). The NOx concentration value can assess the quality of ammonia uniformity in the system. Value 1 represents the NOx distribution minus the deviation of NOx quantity from the outlet and inlet of the SCR system. The value from this equation should be multiplied with 100% of the NOx performance value to identify the percentage NOx conversion value in the SCR system.

## 4. The Experimental Model and the Parameter Conditions

In this study, the experimental setup used a six-cylinder, four-cycle diesel engine that was water-cooled with natural aspiration. The operating condition was restricted to 1000 rpm because the performance maximum of this type of diesel engine is 1200 rpm. At 1000 rpm, the engine can produce a mass flow of 513 kg/h, yielding NO$x$ of 1330 ppm. This parameter followed previous research [1,6,7,15], and it can be used to compute the reduction of NO$x$ in the system. The heavy duty engine specification of this experiment are listed in Table 2. The experimental setup and schematic diagram are shown in Figures 15 and 16.

**Table 2.** Heavy-duty diesel engine specifications.

| Engine | D6CC |
| --- | --- |
| Manufacturer | HYUNDAI |
| No. of cylinders | 6 |
| V-Angle (0 = In-line, 60; 90; 180 = Boxer) | V-60 |
| Strokes (2 or 4) | 4 |
| Type (Otto, Diesel, Turbodiesel) | Diesel |
| Rated power P (kW) | 338/1800 rpm |
| Rated torque M (Nm) | 400/1000–1800 |
| Min. speed nmin (rpm) | 1000 |
| Max. speed nmax (rpm) | 1800 |

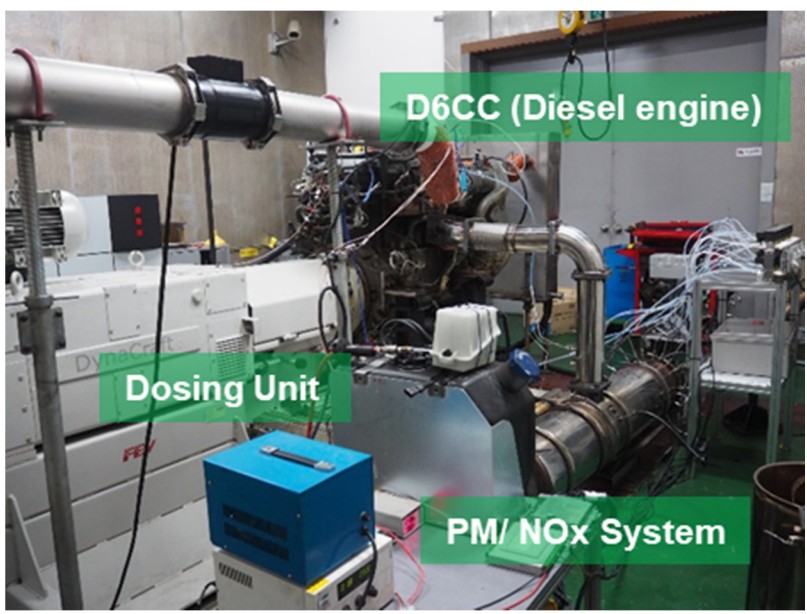

**Figure 15.** Experimental test and engine measurement setup.

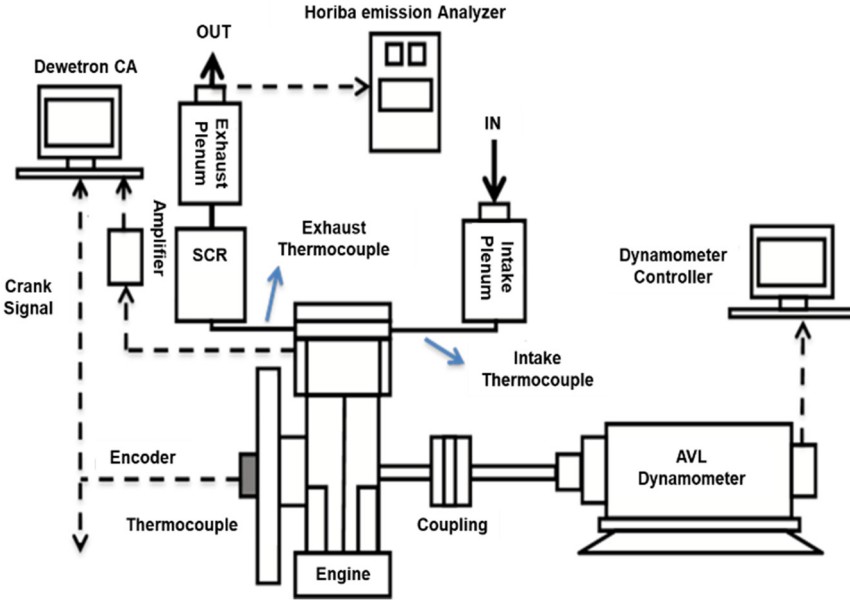

**Figure 16.** Measurement setup and schematic diagram of the test engine.

The UWS temperature used in this system was 263 °C, with a urea flow rate of 1319 mL/h; the ambient temperature was 298 K, and the exhaust gas temperature was 686 K [7,16,17]. The UWS injection specifications are listed in Table 3. The commercial catalyst of the Hyundai D6CC with vanadium (V-TiO$_2$) SCR, a cell density of 300 cpsi and cordierite substrate material was used in this study.

**Table 3.** Value of ammonia and exhaust gas.

| Engine Operation Points | Engine Speed = 1000 rpm | |
| :---: | :---: | :---: |
| | Experimental Conditions | Unit |
| Engine operating point | 1000/200 | rpm/Nm |
| Exhaust mass flow rate | 513 | kg/h |
| Injector inlet/SCR inlet temperature | 371 | °C |
| NO$_x$ quantity (NO) | 1330 (1285) | ppm |
| Downstream SCR line back pressure | 24 | mbar |
| NO$_x$ flow rate | 1083 | g/h |
| AdBlue flow rate | 1319 | mL/h, NSR = 1.0 |
| O$_2$, Volume | 8 | % |
| CO$_2$, Volume | 9.3 | % |
| H$_2$O, Volume | 9.3 | % |

The mixing process between NO$x$ and ammonia particles occurred from the injector to the catalytic converter. That process was assisted by the mixer fan in the system. The gas analyzer (Horiba MEXA-7100 DEGR) was connected on the outlet of the commercial SCR system from the engine. This process could easily analyze the NO$x$ reduction quality in the system.

## 5. Conclusions

An investigation with two types of urea injectors in an SCR system for heavy-duty diesel engines was conducted. The experiment and simulations were performed to improve ammonia quality from

the urea injector. The quality of urea injector performance will assist urea decomposition in the SCR system. The experiment and simulation were based on a 12,000 cc heavy-duty diesel engine operating at 1000 rpm. This engine can produce 513 kg/h of mass flow and a 371 °C exhaust temperature. The parameters used in this study can be a benchmark to obtain similar results at different engine capacities and engine speed conditions.

In this study, I-type and L-type injectors had a fairly uniform pattern of gas distribution, but the ammonia gas ratio from the I-type injector was better than that from the L-type injector in the SCR system. The I-type urea injector used in this system produced 5% more ammonia than the L-type injector. That value shows that the I-type injector was superior in the urea decomposition process to the L-type urea injector, reducing the possibility of urea solid deposition in the system.

These results were validated through additional experiments to describe the quality of the two types of urea injectors. Based on experiments with a gas analyzer at the catalyst inlet, the I-type urea injector produces a larger ammonia quantity than the original urea injector (L-type). However, the L-type urea injector produces greater ammonia uniformity than the I-type urea injector. These result occur because the position of the L-type injector assists the distribution of ammonia into the catalyst surface. Based on that different results, it can be concluded that the I-type urea injector is better for ammonia production, though the L-type urea injector is better at distributing ammonia particles in the system. These results can also be applied to determine the quality NO*x* conversion efficiency in heavy-duty diesel engines. With higher ammonia quantity, the I-type injector has higher NO*x* efficiency results than the L-type injector. However, the ammonia quantity and quality still need improvements and modifications in order to reduce catalyst load and to achieve a high-quality SCR system.

**Author Contributions:** Conceptualization, M.K.A.W.; methodology, M.K.A.W.; software, M.K.A.W.; validation, M.K.A.W.; formal analysis, M.K.A.W.; investigation, M.K.A.W., K.O. and O.L.; resources, M.K.A.W.; data curation, M.K.A.W., K.O. and O.L.; writing—original draft preparation, M.K.A.W.; writing—review and editing, M.K.A.W., K.O. and O.L.; visualization, K.O. and O.L.; supervision, K.O. and O.L.; project administration, K.O. and O.L.; funding acquisition, K.O. and O.L. All authors have read and agreed to the published version of the manuscript.

**Funding:** This research is financially supported by the Global Top Environmental Technology Development Project of the Korea Environmental Industry and Technology Institute (RE202001110, Development and Demonstration of simultaneous PM and NOx reduction system of military vehicles and RE2016001420002; Development of the PM·NOx purifying system and the core technology; Shipbuilding and Offshore Industry Core Technology Development Business by the Ministry of Trade, Industry and Energy (MOTIE, Korea) [Development of Low Print Point Alternative Fuel Injection System for Small and Medium Vessel Enfines for Ships Hazardous Emission Reduce]. (20013146).

**Acknowledgments:** This work was supported by a research program in the Department of Mechanical Engineering (Generation Fuel and Smart Power Train Laboratory), University of Ulsan, Republic of Korea. This research is financially supported by the Global Top Environmental Technology Development Project of the Korea Environmental Industry and Technology Institute (RE202001110, Development and Demonstration of simultaneous PM and NOx reduction system of military vehicles and RE2016001420002, Development of the PM·NO*x* purifying system and the core technology.

**Conflicts of Interest:** The authors declare no conflict of interest.

## Nomenclature

| | |
|---|---|
| *RANS* | *Reynolds Average Navier Stokes* |
| *CFD* | *Computational Fluid Dynamics* |
| *UWS* | *Urea Water Solution* |
| *SCR* | *Selective Catalyst Reduction* |
| $H_2O$ | *Hydrogen* |
| $NO_x$ | *Nitrogen Oxide* |
| $N_2$ | *Nitrogen* |
| $O_2$ | *Oxygen* |
| *K* | *Kelvin temperature* |
| *UI* | *Uniformity Index* |

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
