# Peer review of "Investigation of Urea Uniformity with Different Types of Urea Injectors in an SCR System"

_catalysts, doi:10.3390/catal10111269_

Round 1

Reviewer 1 Report

The revised manuscripts was improved based on the reviewers’ comments. There are still some issues need to address:

  1. What is the SCR catalyst in this work? Cu-Zeolite, Fe-Zeolite or hybrid catalyst? What is the support material, SAPO-34, SZZ-13, ZSM-5 or others? If it the commercial catalyst, SCR catalyst information should be offered.
  2. What is the urea distribution when the exhaust gas is not as strong as 513 kg/h?
  3. Will the urea clogging occur at the injector nozzles with time going on? Which injector is easier to be clogged considering the temperature distribution near nozzle area?
  4. In my last review, the grammar issue in the title was pointed out, but it was still not corrected. Author needs to check the paper carefully.

Author Response

Reviewer 1

The authors are very grateful with the comments and recommendations in order to improve the quality of this paper. We agreed and accepted the reviewer’s suggestion. The manuscript is revised carefully according to the reviewer’s comments and detailed corrections are listed below point by point. Responses to the reviewer’s comments are as follows

  1. What is the SCR catalyst in this work? Cu-Zeolite, Fe-Zeolite or hybrid catalyst? What is the support material, SAPO-34, SZZ-13, ZSM-5 or others? If it the commercial catalyst, SCR catalyst information should be offered.

Answer:

The authors would like to thank for reviewer’s comment and suggestion in order to improve the quality of this paper. We use the commercial catalyst from Hyundai D6CC with Vanadium (V-TiO2) SCR, with cell density 300cpsi and Codierite substrate material.

The information about catalyst explained at line 196 to 198.

“The commercial catalyst from Hyundai D6CC with Vanadium (V-TiO2) SCR, with cell density 300cpsi and Cordierite substrate material used in this study.”

  1. What is the urea distribution when the exhaust gas is not as strong as 513 kg/h?

Answer:

The authors would like to thank for reviewer’s comment. The urea distribution in the system based on the quantity of NOx. When the exhaust gas is not as strong as 513 kg/h the NOx sensor will inform the controller to decrease the urea injection quantity. However, the operating condition in this study was restricted to 1,000 rpm and produce 513kg/h. so we didn’t do the experiment below that condition.

The information about parameter was explaining at line 183 to 188.

“In this study, the experimental setup used a six-cylinder, four-cycle diesel engine, water-cooled with natural aspiration. The operating condition was restricted to 1,000 rpm because the performance maximum of this type of diesel engine is 1,200 rpm. At 1,000 rpm, the engine can produce a mass flow of 513 kg/h, yielding NOx of 1,330 ppm. This parameter followed previous research [1], [6], [7], [14], and it can be used to compute the reduction of NOx in the system. The experimental setup and schematic diagram are shown in Figures 6 and 7”

The NOx sensor position.

  1. Will the urea clogging occur at the injector nozzles with time going on? Which injector is easier to be clogged considering the temperature distribution near nozzle area?

Answer:

The authors would like to thank for reviewer’s comment. Yes, the injector nozzles can be clogging with time. The L-type urea injector easier to be clogged in this study. Because the position of injector spray was in the middle of pipe. That injector clogging will reduce the quantity of urea distribution in the system. However, we didn’t analyst that situation in this study; but we will deeply analyst that phenomena in the next study.

The picture information regarding the clogging phenomena.

  1. In my last review, the grammar issue in the title was pointed out, but it was still not corrected. Author needs to check the paper carefully.

Answer:

The authors would like to thank for reviewer’s comment and suggestion. Yes, we will carefully to check the quality of this paper; and the entire manuscript has been checked and revised carefully with professional English checker.

Reviewer 2 Report

 The current revised version of the manuscript is suitable for publication in catalysts. 

Author Response

Reviewer 2

The current revised version of the manuscript is suitable for publication in catalysts.

Answer:

The authors are very grateful with the comments and recommendations in order to improve the quality of this paper. Thank you very much.

Reviewer 3 Report

The revised version can be published, as the required changes have been carried out.

Author Response

Reviewer 3

The revised version can be published, as the required changes have been carried out.

Answer:

The authors are very grateful with the comments and recommendations in order to improve the quality of this paper. Thank you very much.

Round 2

Reviewer 1 Report

The authors have addressed the remained issues and revised the manuscript accordingly based on reviewers' comments. The manuscript quality has been largely improved. Therefore, I suggest this manuscript be accepted in the present form. 

This manuscript is a resubmission of an earlier submission. The following is a list of the peer review reports and author responses from that submission.

Round 1

Reviewer 1 Report

The manuscript “The investigation of urea uniformity with difference type of urea injector in SCR system” studied L-shape and I-shape urea injectors and the effects of injector shapes on the de-NOx performances. I read through the whole manuscript and find the following main problems:

  1. This paper is focusing on the study of urea injectors with different shapes. This research topic is not within the scope of Catalysts journal, which is for the insightful understandings of catalyst materials and reactions.
  2. The goal of this research is not significant clear in the introduction part.
  3. Too many grammar errors in the text, even the grammar in title should be corrected.

Therefore, I could not recommend this paper to be considered to published in Catalysts journal.(Unmatching with the journal scope is the main reason).

Reviewer 2 Report

The manuscript is interestingly written and concerns interesting studies with significant application input. However, I have some comments that should improve the manuscript:

- Authors should write down in the manuscript possible reactions that occur during the conducted research.

-Which means UI on fig. 15 ?

- Are there any known types of urea injectors besides those used in the studies described in the manuscript? If so, the authors should justify their choice of urea injectors for the conducted experiments

- More references should be cited in Introduction section to have a better overview of the research topic.

- The manuscript  must be corrected for editing errors, e.g. missing subscripts line 185 or 220.

Reviewer 3 Report

The manuscript entitled: 'The investigation of urea uniformity with difference type of urea injector in SCR system' shows a comparison of two types of urea injectors. The manuscript present enormousness lacks. Introduction does not provide enough information and should be rewritten. The results section should be revisited and improved, especially the discussion. The conclusion section should be improved. Several typos mistake are present. Overall, the manuscript in the present form should be not considered for publication.

Reviewer 4 Report

Review Report for Catalysts manuscript n. 972314

The investigation of urea uniformity with difference type of urea injector in SCR system

The topic of the paper is interesting and matches the scopes of the journal. The paper reports an interesting experimental study coupled with CFD simulation of the SCR of NOx in a diesel engine.  The manuscript deserves to be published in Catalysts after major revision.

Abstract

The Abstract should report quantitative results, just few lines about them. Instead, only generic assumptions were made, like "The results in this study show the temperature phenomena, gas distributions, ammonia uniformity patterns, and NOx reduction efficiency of the two types of urea injectors. Investigations on the quality of ammonia uniformity at the catalyst inlet showed the chemical mechanisms between quantity of ammonia concentration distribution and NOx mixing process."

  1. Introduction

The state of the art is rather poor, with 7 papers cited only. This paragraph has to highlight the problems still unsolved of a certain field.

The statement "This study assesses ammonia uniformity and NOx reduction efficiency to elucidate the urea decomposition process and solid deposition phenomena. Our experiments involved..." should be moved to the end of the paragraph, where the authors explain what they have done.

The innovation of the study and the findings besides the state of the art has to be indicated at the end of this paragraph as well.

Was the NOx concentration measured upstream and downstream the SCR catalysts? If so, please provide the conversion yields during the tests.

  1. Conclusions

Please provide quantitative results of your work.